# Role of Infodemics on Social Media in the Development of People’s Readiness to Follow COVID-19 Preventive Measures

**DOI:** 10.3390/ijerph19031347

**Published:** 2022-01-25

**Authors:** Bireswar Dutta, Mei-Hui Peng, Chien-Chih Chen, Shu-Lung Sun

**Affiliations:** 1Department of Information Technology and Management, Shih Chien University Taipei Campus, Taipei 10462, Taiwan; 2Institute of Information Management, National Yang-Ming Chiao Tung University, Hsinchu 300044, Taiwan; clare4260@gmail.com (M.-H.P.); slsun223388@gmail.com (S.-L.S.); 3Institute of Information Management, Minghsin University of Science and Technology, Hsinchu 300044, Taiwan; profchencc@gmail.com

**Keywords:** misinformation, social media, COVID-19, pandemic, health behavior, government

## Abstract

Unparalleled levels of misinformation have contributed to widespread misunderstandings about the nature of the coronavirus, its cure and preventative measures. Misinformation crosses borders rapidly with the help of social media, and this phenomenon is constantly increasing. Thus, the current study proposes a research framework to explore how citizens’ trust in government and social media influences their readiness to follow COVID-19 preventive measures. Additionally, the role of a health infodemic was explored in perceptions and relationships among factors influencing an individual’s readiness to follow COVID-19 preventive measures with data collected from 396 participants in Taiwan. The findings indicate citizens’ trust in social media (TRSM), attitude (ATT), perceived benefit (PBT), personal innovativeness, and how peer referents positively influence their readiness. However, the relationship between citizens’ trust in the government (TRGT) and their readiness to follow COVID-19 preventive measures (INT) is not statistically significant. The current study also explores the negative moderating effect of health infodemics on the relationship between TRSM and INT, TRGT and INT, ATT and INT, PBT and INT. Thus, the Taiwanese government must consider the current study’s findings to develop attractively, informed, and evidence-based content, which helps its citizens improve their health literacy and counter the spread of misinformation.

## 1. Introduction

Coronavirus (COVID-19) was initially found in December 2019 at a wholesale seafood market in Wuhan, China [1]. It is an exceptionally infectious disease. Following the onset of Middle East respiratory syndrome coronavirus (MERS-CoV) in Saudi Arabia and severe acute respiratory syndrome coronavirus (SARS-CoV) in China, the COVID-19 pandemic is considered the third of its kind, which threatens human civilization globally [2]. The severe spread of coronavirus (COVID-19) has posed sensational threats to public safety, the economy, and education globally. Nearly all workplaces, educational institutions, and public spaces have been entirely shut down [3].

Virtually every country has taken numerous prevention measures to save its citizens from the COVID-19 pandemic. One of the standard measures, complete lock-down, is implemented in several countries. Other than a complete lock-down, “#StayAtHome” is considered a fruitful measure to prevent the dissemination of COVID-19 [3].

Currently, several countries have started a vaccination program for their citizens as it helps the body fight against viruses and improves the immune system to respond to an invader [4]. Vaccines from different companies train the body to fight against the invader differently and are named based on how it works [4]. For example, Pfizer and Moderna vaccines are referred to as “mRNA vaccines”, and the AstraZeneca and Johnson & Johnson vaccines are considered “adenovirus vaccines” [5]. However, the vaccination rate is still below the expected level for several reasons. Thus, the only counteractive option is to follow the preventive measures implemented by governments to prevent the further spread of the COVID-19 pandemic [6].

Taiwan’s government, similar to its counterpart, introduced several measures to prevent the further spread of the coronavirus. For example, the government has encouraged citizens to wash their hands at a regular time interval, maintain social distance in public areas, and measure individuals’ temperatures regularly in crowded places such as convenience stores, educational institutions, and dining halls. Using masks outside the household became mandatory, along with registering every citizen’s contact information, encouraging citizens to clean and disinfect the living environment, etc. [7]. Taiwan’s government dynamically accelerated the vaccination program after facing a critical outbreak in April 2021 [7]. Several ministries such as the Ministry of Education, the Ministry of Economic Affairs, the Ministry of Labor, and the Ministry of Health and Welfare agreed that teachers at junior high schools, kindergartens, community colleges, cram schools, nurses and physicians, etc., must be prioritized and fully vaccinated by Jan. 1, 2022. A total of 78.74% of the population have received two doses, and 63.8% have received one short of vaccine [7]. Taiwan’s Government also restricted its border. Each traveler entering the country must undergo mandatory quarantine and take polymerase chain reaction (PCR) tests, irrespective of their vaccination status [7].

Although implemented measures and vaccination led to early success, positive cases were always under control. However, the number of reports on hesitancy in following the citizens’ measures started increasing. Even most Taiwanese citizens have no prior experience and are not familiar with these sudden changes. Thus, citizens’ readiness to follow such measures remains unidentified [3]. Additionally, Taiwanese citizens need time to deal with these sudden changes because these changes could continue for an extended period and emerge as a new way of living in the post-COVID-19 period. However, although citizens need to follow these sudden changes due to the situational perspective, the gap in their psychological cognition may lead to differences in behavior effect and behavioral process [8]. Thus, to understand this gap, the current study explores citizens’ psychological, cognitive functions during this pandemic outbreak.

Social media has emerged as the most active communication tool in the current digitalization age and plays a vital role in increasing citizens’ awareness and attitude towards definite concerns. Administrations also use it to reach mass audiences more quickly and communicate with them more effectively. However, a key challenge confronting social media is that the information generated and exchanged is always not dependable [8]. Misinformation or an infodemic is the rapid spread of incorrect or inaccurate information (intentionally or unintentionally) on social media [9]. Health infodemics cause a mess, create negative emotions and misinterpretation, and harm fellow citizens during the pandemic. Furthermore, the circulation of misinformation can harshly impair the effectiveness and efficient usage of information drawn on social media [10]. Chen and Sin [8] explored the rapid spread of misinformation as one of the top ten considerations faced by governments of different countries. 

To better cope with citizens’ readiness to follow COVID-19 preventive measures, it is also essential to consider the peer referent and personal innovativeness issues as literature explored opinions of friends and family members influenced an individual’s intention to accept a new system [11,12]. Additionally, the better innovativeness of an individual develops a more positive acceptance behavior [13,14]. However, the findings reported in these investigations are still inconsistent and indecisive. Thus, the current study incorporates peer referent and citizens’ innovativeness for better understanding and predicting citizens’ perception to follow COVID-19 preventive measures. The purpose of the current study is to explore the factors influencing citizens’ readiness to follow COVID-19 preventive measures during the COVID-19 pandemic. We employ two factors based on existing literature developing a construct called citizens’ trust. Additionally, citizens’ attitude and perceived benefit are also incorporated to comprehend citizens’ perceptions of following COVID-19 preventive measures. 

## 2. Research Context, Hypotheses, and Methods

### 2.1. Health Infodemic

The term “infodemic” [15,16] indicates an extreme volume of information that is typically unreliable, spreads rapidly, especially during the administration of a virus epidemic [17,18,19], as it has the potential to accelerate the pandemic interaction by affecting and dividing social response [20]. In conclusion, the term “infodemic” denotes too much fabricated or misinformation during a virus epidemic in digital and physical circumstances. 

Misinformation is generally prevalent during the early phase of pandemics, such as during the HIV epidemic. The effects of conspiracy theories, rumors, and misinformation are still visible in these areas today [21]. This was primarily emphasized by the Mbeki South African government’s denial of HIV in the early 2000s and their infamous refusal of the proposal involving the efficacy of HIV medication [21]. The government promoted the unconfirmed use of herbal remedies, incorporating garlic, beetroot, and lemon juice for AIDS treatment [21] as an alternative solution. This disreputable action severely affected pregnant mothers’ children and cost more than 300,000 lives [21].

As COVID-19 turns into a full-fledged public health catastrophe, several theories concerning the virus’s origin have been instigated on the internet [21,22]. Despite all efforts of scientists from several countries, the theories are continually originating and gaining popularity on the internet. Moreover, the high rate of COVID-19 publications indicates the vital effort by the scientific community to understand the virus, the way of transmission, and COVID-19 disease pathology in a better way, unquestionably helping citizens learn more [22]. However, there is a concern that such a great rate of inclusion in the literature may be related to unintentional blunders, inaccuracies, and differing degrees of misinformation [22]. 

Information can be broadcast more quickly with the rapid digitization of society and the development of social media [23]. It could assist in blocking up information cavities more quickly. However, it could also intensify the flow of detrimental information [22,23]. An infodemic regarding COVID-19 can amplify or pull out epidemics, whereas citizens are indeterminate about what they should do to be healthy [21]. 

The public health crisis emerging from the coronavirus (COVID-19) also started to feel the effect of misinformation [21]. Misinformation has spread far and wide, drowning out credible sources of information [22]. Information from uncredited sources has clouded the preliminary details on reducing transmission and exposure to the virus [23]. The spread of false information masks credible sources, creating further confusion in public, ultimately leading to worse and inefficient information concerning virus transmission [24]. It also creates distrusts among citizens towards health authorities and demoralizes their positive health behaviors. Many begin to believe that risk-taking behaviors will help to improve well-being [24].

Infodemic management is the systematized practice of risk and indication-based analysis and methodologies. It helps to lessen the effect of misinformation on citizens’ health performances during health occurrences. Infodemic management intends to support moral health behaviors through four types of behaviors, such as 1. Eavesdropping of public anxieties and queries; 2. they endorse considerate threat and health expert recommendations; 3. developing resiliency against misinformation; 4. participating and endowing citizens to take affirmative actions [25]. 

The COVID-19 epidemic has exposed the critical influence of this innovative information atmosphere. The information diffusion could strongly influence citizens’ actions and reduce the administrations’ efficiency to take corrective actions. The effect of these false opinions can be so transferrable that they can influence governmental policy, which has the potential to be fatal [21,23]. In the face of a pandemic, governments need to be transparent. They should communicate clearly and provide trustable information to the public as confusion leaves citizens unprepared to combat a public health crisis [22,24,25]. Additionally, it is dangerous for governments to politicize this pandemic. 

A critical study challenge is determining how citizens search for or avoid healthcare-related information and how these determinations affect their healthcare-related actions [26]. The news phase, dominated by the disintermediated dissemination of information, modifies the approach information is paid out and commented on.

### 2.2. Role of Social Media during COVID-19 Pandemic

Social media has become a permanent component of the public health prospect [27,28,29]. The Internet offers an increasingly vital resource for citizens, from online meetings to web-based training for smoking cessation, weight loss, etc. The Internet provides an increasingly essential resource for citizens concerning health issues [30]. However, most of these attempts focus on administrative implements for delivering better facilities to citizens. Evenly significant social media habit has come from the development of peer-driven health groups [31].

Social media platforms offer straightforward access to an unparalleled content volume and intensify rumors and dubious information. Algorithms intermediate and simplify content promotion and information distribution [27]. This swing from the outdated news pattern intensely influences the creation of social sensitivities [28] and encloses narratives. It impacts policy-making and the development of public discussion [29,30], specifically while matters are provocative [31]. Users are inclined to obtain information following their stances [32] while overlooking nonconforming information [33] and shaping differentiated groups around collective narrations [34]. Moreover, misinformation might boom while divergence is high [35]. The literature indicated that fake news and imprecise information might diffuse quicker and more inclusive than fact-based information [31]. 

Exploring the effect of social media to improve the awareness of differing concerns has also been considered in COVID-19. Thus, a considerable research challenge is determining how information spread by social media influences citizens’ healthcare-related actions. The complications of contemporary infodemics are unquestionably being presumed from several perspectives. It comprises the dynamics of counterfeit information to the concerns of diffusion of infodemics [30,36].

### 2.3. Research Hypotheses

#### 2.3.1. Trust and Citizens’ Perception to Follow COVID-19 Preventive Measures

Trust is an association between trustor and trustee. It has entranced academicians of different fields such as sociology, political science, economics, etc. Their investigations concentrated on how trust influences administrations and civilization. Several academicians started to pay attention to the influence of trust on consumers’ adoption intention due to the rapid digitalization. 

Citizens’ trust in government refers to citizens’ confidence that government can take the appropriate steps to provide truthful information to satisfy citizens’ interests. From the healthcare perspective, citizens with a better degree of trust in government feel less ambiguity. They believe that the provided information is trustworthy and will improve their effectiveness in managing their health behavior [37]. Correspondingly, citizens’ trust in the government improves their attitude as they consider that it adequately developed and maintained an information system [38]. The literature explored that trust in government positively influences citizens’ motivation to follow positively and significantly affects citizens’ attitude, which further improves their motivation to follow [37,38]. Kapoor et al. [39] found a positive relationship between trust in government and consumers’ sustainable motivation to use social media, which influences citizens’ intention to adopt a definite social media. Yildiz [40] studied employees’ organizational behavior and found that employees’ trust in an organization supports them in taking positive attitudes and actions. 

Trust in social media is regarded as an institutional-basis trust, which denotes individuals’ understanding of social media assurance strategies and guidelines that feel safe. Social media is the envoy for delivering information to citizens. Social media trust is constantly recognized as a critical interpreter of citizens’ perception of COVID-19 preventive measures. The literature explored how trust in the Internet positively influences citizens’ readiness to follow COVID-19 preventive measures [41,42,43]. These investigations explored that citizens trust that social media is dependable and safe and might support bug-free reliable information. Citizens’ positive perception improves their attitude to use, further improving their readiness to follow COVID-19 preventive measures. Thus, we hypotheses:

**H1.** 
*Trust in the government positively affects citizens’ readiness to follow COVID-19 preventive measures.*


**H2.** 
*Trust in social media positively affects citizens’ readiness to follow COVID-19 preventive measures.*


**H3.** 
*Trust in the government has a positive effect on attitude.*


**H4.** 
*Trust in social media has a positive effect on attitude.*


Individuals’ trust in social media is regarded as one of the most significant psychological determinants influencing their online behaviors [44]. The literature indicated a positive relationship between trust in social media and perceived benefit [45,46]. Individuals with a greater degree of trust in social media feel less ambiguous and consider that information will help them understand the concern in a better way [44]. Correspondingly, individuals’ trust in government can significantly consider the perceived benefit through cost-benefit analysis. The literature explored citizens’ trust in government significantly influenced their perceived benefit [45,47]. The notion of trust in the current investigation is multidimensional. It primarily covers citizens’ trust in institutions that offer the services and the social media or technology that provide services. Thus, we hypothesize: 

**H5.** 
*Trust in the government has a positive effect on perceived benefit.*


**H6.** 
*Trust in social media has a positive effect on perceived benefit.*


#### 2.3.2. Attitude and Citizens’ Readiness to Follow COVID-19 Preventive Measures

Dutta et al. [48] stated that attitude is an individual’s definite characteristics that outline either positive or negative behavior and replicate sensation and information to specific subjects. In their study on adopting healthcare technology, Sun et al. [13] theorized that attitude plays an essential role in impacting an individual’s behavioral intention. Hwang et al. [49] explored physicians’ attitude towards healthcare technology that influenced their intention to use electronic medical records (EMR). Venkatesh et al. [50] reported that attitude towards information technology (IT) is the second most significant predictor regarding physicians’ intention to use telemedicine services. Thus, we hypothesize:

**H7.** 
*Attitude has a positive effect on citizens’ readiness to follow COVID-19 preventive measures.*


#### 2.3.3. Perceived Benefit and Citizens’ Readiness to Follow COVID-19 Preventive Measures

Perceived benefit denotes individuals’ benefits if they accept self-consciousness and treatment information about COVID-19 on the Internet [51]. The encouraging knowledge attained by individuals from obtaining and taking health information behavior will endorse the inclusive value observation of this behavior. Gong et al. [52] explored the idea that individuals generally consider the efficiency of behaviors through cost-benefit assessment while accepting well-being behaviors. The consideration of benefits offered a more privileged accomplished path [53,54]. Building on conclusions by Gong et al. [52], we examined how perceived benefits are related to behavioral intention. Thus, perceived benefits were applied to the current research model as a determinant that evaluates citizens’ perception of the effectiveness of COVID-19 preventive measures, which further reinforces citizens’ intention of continuing to follow COVID-19 preventive measures. 

**H8.** 
*Perceived benefit has a positive effect on citizens’ readiness to follow COVID-19 preventive measures.*


#### 2.3.4. Personal Innovativeness and Citizens’ Readiness to Follow COVID-19 Preventive Measures

Individuals are dissimilar in their tendency to follow novel regulations [14]. Generally, innovation diffusion investigations identified that highly innovative individuals are active information explorers. They can cope with more significant uncertainty and develop more constructive behavior toward adoption [13,55]. Chen et al. [56] stated personal innovativeness is an individual’s readiness to try out definite innovative technology and regarded it as the significant behavioral intention to follow definite regulation. Sun et al. [13] found personal innovativeness influenced nurses’ behavioral intention to use Tablet PC. Thus, we hypothesize: 

**H9.** 
*Personal innovativeness positively influences citizens’ readiness to follow COVID-19 preventive measures.*


#### 2.3.5. Peer Referent and Citizens’ Readiness to Follow COVID-19 Preventive Measures

Peers have a rational influence and significantly influence adoption behavior in a healthcare environment. Potarca [11] recommended that relationships between peers can predict individuals’ well-being, as references between peers are instinctive. For example, if most of your friends approve of a view, you are pleased to display your accordance. Although an individual’s performance is compared with a friend’s, approval will happen if his/her behavior is acknowledged by that friend [12]. If peers understand an individual’s information, a significant peer referent is created in healthcare behaviors. These referents are related to individual assessment of wellbeing concerning healthcare behavioral development. Sakallaris et al. [57] explored how peer support significantly influences citizens’ perception, motivation, and assurance to follow.

**H10.** 
*Peer referent positively affects citizens’ readiness to follow COVID-19 preventive measures.*


#### 2.3.6. Moderating Effect of Health Infodemic

A better level of trust in government and social media generally improves citizens’ readiness to follow COVID-19 preventive measures. However, health infodemics (HID) reduce citizens’ willingness to follow COVID-19 preventive measures [52]. Thus, the current study theorizes that HID can moderate the relationship between citizens’ trust and readiness toward COVID-19 preventive measures. If citizens have a similar level of trust in government or social media, they are willing to follow COVID-19 preventive measures. However, if they consider a more significant threat due to infodemics during this process, they may stop following COVID-19 preventive measures. Thus, the current study theorizes that HID negatively moderates the relationship between citizens’ trust and readiness toward COVID-19 preventive measures. 

A positive attitude refers to the individual’s perception of a specific behavior. Previous studies validated that citizens observe guidelines, necessities, and strategies while having a positive attitude [13,48,49,58]. On the other hand, infodemics negatively influence citizens’ attitude, influencing them to give away a specific behavior. Thus, the current study considers how HID might negatively affect the relationship between citizens’ attitude and readiness toward COVID-19 preventive measures. 

Perceived benefit is the degree to which an individual could improve through the practice of a specific service [52]. If citizens consider that following regulations benefit them, they show positive intention. However, infodemics negatively influence citizens’ motivation to follow new measures, resulting in quitting the behavior. Thus, the current study theorizes that HID negatively moderates the relationship between perceived benefit and citizens’ readiness to follow COVID-19 preventive measures. 

Thus, we propose the following hypotheses.

**H11a.** 
*Health infodemics (HID) negatively moderate the relationship between TRGT and citizens’ readiness to follow COVID-19 preventive measures.*


**H11b.** 
*Health infodemics (HID) have a negative moderating effect on the relationship between TRSM and citizens’ readiness to follow COVID-19 preventive measures.*


**H11c.** 
*Health infodemics (HID) have a negative moderating effect on the relationship between attitude and citizens’ readiness to follow COVID-19 preventive measures.*


**H11d.** 
*Health infodemics (HID) have a negative moderating effect on the relationship between perceived benefit and citizens’ readiness to follow COVID-19 preventive measures.*


#### 2.3.7. Research Model

Based on the literature mentioned above, a research model is proposed for evaluating the role of trust, attitude, perceived benefit, and health infodemics in citizens’ readiness toward COVID-19 preventive measures (Figure 1).

## 3. Materials and Methods

### 3.1. Measurement and Survey Design

The current study used mixed methodologies to develop and validate the proposed model. The proposed study model development comprised literature review and in-depth interviews with experts of the related subject both from industry and academia. Later, focus group discussions were used to develop conclusive discussion and interpretation of the answers from the empirical investigation afterward. The study model was then empirically verified by administering survey methodology to the research instrument designed for the study. The source of the items is stated in Appendix A. 

The literature indicated demographic features which also affect citizens’ readiness to follow COVID-19 preventive measures. Thus, demographic characters were also included in the questionnaire. The questionnaire was divided into three sections; the purpose of the study was described in the first section. The second section contains multiple-choice items regarding personal information such as gender, age, educational background, and the number of years the responder has used social media; the final section includes items regarding study constructs. The final questionnaire consists of the eight predictors and the demographic information of responders.

### 3.2. The Delphi Method

The Delphi method was used to validate the initial conceptual framework. A total of experts formed an expert panel. Out of 4 panelists, 2 panelists were male and working as assistant professors. Only 1 panelist, female, was working as an instructor at the university. All the panelists have more than 12 years of experience in the field of information science. Another panelist worked in a hospital with over ten years of experience as a senior health informatics associate. Three experts have received their Ph.D. degrees already, and another one will receive her Ph.D. degree shortly. The age distribution of the experts was between 40–52 years (3 of them belong to the 40–50 group and 1 belongs to 45–50). A significant concern to ask those experts to serve as the panel members is their expertise areas that are much more related to information science. After two rounds of expert meetings, a research framework and items were proposed for a pilot study. We used a 5-point Likert scale, with a measure of strongly disagree, disagree, not sure, agree, and strongly agree. Each round of expert panel meetings was separated by three weeks to avoid the memory effect. The questionnaire modifications were carried out based on expert opinions and relevant literature reviews to increase the content validity. 

After receiving the questionnaire from experts, the mean value, standard deviation, and internal consistency were verified, respectively. Chronbach’s α values in the first round ranged from 0.745 to 0.972. The expert panel suggested three modifications: (1) simplifying the questions that are similar and difficult to understand; (2) systematics presentation of the questions; (3) including one item to readiness toward COVID-19 preventive measures. Furthermore, two items were deleted due to the redundancy of the items. Therefore, 25 items were sent for round two of the Delphi method.

After the second round, Chronbach’s α value is 0.814 in pathways, and others ranged from 0.864 to 0.985. The average of the importance of each factor turns out between 3.82 to 5.00, with a standard deviation between 0.5 to 1.5. Based on the suggestions from the second-round meeting, items of attitude, perceived benefit, and perceived referent remain the same as the proposed initial research framework. However, items were increased from 4 to 5 for health infodemics and 2 to 3 for trust in government and social media.

### 3.3. Data Collection

The target population for the current study was Taiwanese citizens. Researchers currently employ different survey technologies, Sentiment analysis, which uses advanced artificial intelligence technologies such as Natural Language Processing (NLP), text analytics, and data science to identify, extract, and explore subjective information in positive, negative, or neutral manners [59]. We used the convenience sampling method; because it is cost-effective and has been extensively used in information system (I.S.) research [60] as well as allowing the researcher to obtain primary data and trends regarding the study without the complications of using a randomized sample [61]. An online survey was conducted to collect data. Respondents were also informed of their rights to be withdrawn from participation at any time during the study. 

### 3.4. Demographic Information of Participants

Data for the current study were collected through structured questionnaires administered to respondents. A total of 405 responses were returned. However, 9 responses were unable to be used due to incomplete responses, missing data, etc. Thus, 396 responses were finally used to validate our proposed model. Figure 2, Figure 3, Figure 4 and Figure 5 indicate how the current study respondents differ by gender, age, educational qualification, and social media experience, respectively. According to national data of Taiwan, the ratio of females and males is 50.3% to 49.7% [62], and 71.35% of the population’s age is between 15 to 64 years [62]. However, our sample’s ratio of males and females is 52% and 48%, respectively. The majority of respondents (78%) are in the age group between 18–45, indicating that the sample is unbalanced in terms of gender and age. Most of them (64.2%) had a college degree. In conclusion, the respondents were highly educated, mature enough, and familiar with using social media.

## 4. Data Analysis

Data analysis was conducted in three steps. First, testing the measurement model’s convergent validity and discriminant validity, and subsequently testing research hypotheses and the structural model. The third step examines the moderating effect of HID on the relationships between TRGT, TRSM, ATT, PBT, and citizens’ readiness toward COVID-19 preventive measures.

### 4.1. Descriptive Statistics and Correlation

Means and standard deviations of ATT, TRGT, TRSM, PBT, PIIT, PRT, HID, and INT are reported in Table 1, which indicate that citizens have a greater extent of attitude and trust and low perception concerning health infodemics, and better consideration to follow COVID-19 preventive measures. The standard deviations (≤1) further lay down citizens’ considerations on the comparatively uniform sub-factors, and their attitudes are reasonably consistent.

The correlation coefficient for each pair of determinants is demonstrated in Table 2. Each correlation coefficient significantly correlated with each other, apart from health infodemics. HID negatively correlated with ATT, PBT, PIIT, PRT, and INT. These findings indicate that all related factors could be further evaluated using SEM and regression.

Correspondingly, the current investigation also explores common method bias (CMB), as the data of all the constructs (a questionnaire) instigated from the same respondents. In the current study, several practical and statistical analyses were used. Procedures were used regarding procedural CMB matters, effective scales, simple language, etc. Concerning statistical issues, Herman’s single factor scores were verified. The total variance for a single factor is 31.627%, lower than the recommended value of 50% [63]. Thus, the finding points out data has no concerns of CMB.

### 4.2. Structural Equation Model

The current study employs a structural equation model (SEM) using AMOS 22 to analyze the causal relationships of the proposed model. We determined each variable using a 5-point Likert scale; the collected data are continuous. Therefore, the SEM with maximum likelihood (ML) assessment is apposite for the current kind of collected data. Thus, the current study uses confirmatory factor analysis (CFA) to evaluate the reliability and validity of the proposed model. Secondly, the current study uses a structural model to evaluate the influence of ATT, TRGT, TRSM, PBT, PIIT, PRT, and INT. 

#### 4.2.1. Measurement Model

CFA demonstrates that values of fit indices: CMIN/DF = 2.687, GFI = 0.946, CFI = 0.982, IFI = 0.968, and RMSEA = 0.064, indicates good model fit. The reliability is calculated using factor loadings that should be 0.7 or above [64]. The values of all factors are above the recommended value of 0.7. Thus, we can consider data met the reliability. Construct reliability was analyzed by Cronbach’s alpha and composite reliability, presented in Table 2. Cronbach’s alpha of each construct ranged from 0.78 to 0.92, which is above the suggested value of 0.70 [64]. CR values of the latent factors are above 0.70 [64], suggesting reasonable construct reliability and consistency for the measurement items of each construct. 

The average variance extracted (AVE) of each construct should be surpassed the variance because the measurement error of that construct (AVE should be exceeded 0.50) is used to measure the convergent validity of the scales [65]. As Table 2 confirms, the constructs’ AVE values range from 0.64 to 0.81, thus meeting the condition for convergent validity.

#### 4.2.2. Structural Model Testing

Table 3 reports findings of SEM through the path coefficients, critical ratios (CR), probability values (*p*), and hypotheses testing results. The values of fit indices, CMIN/DF = 4.748, GFI = 0.94, CFI = 0.945, IFI = 0.941, and RMSEA = 0.072, validate good model fit. As Table 3 presents, the path coefficients between TRGT and INT are not statistically significant. Thus, H1 is not supported, whereas TRSM has a positive and significant influence on citizens’ readiness toward COVID-19 preventive measures. Thus, H2 is supported. It shows trust in social media plays an important role in following COVID-19 preventive measures as social media is a key source of information.

Similarly, TRGT and TRSM are positively and significantly related to ATT. Henceforth, H3 and H4 are supported. The positive associations also look relatively coherent, as citizens’ better trust levels in government and social media can improve citizens’ attitude toward following COVID-19 preventive measures. Similarly, the findings of path analysis also support H5, H6, H7, H8, H9, and H10. It indicates citizens’ ATT, PBT, PIIT, and PRT are positively and significantly associated with citizens’ readiness toward COVID-19 preventive measures. Citizens’ positive attitude suggest that they are entirely ready to follow COVID-19 preventive measures. Thus, H7 is supported. Citizens’ higher innovativeness indicates that they positively perceive COVID-19 preventive measures and are prepared to follow them. Similarly, a better perception of benefit also improves citizens’ readiness to follow COVID-19 preventive measures. Finally, the more citizens talk about preventative measures with their peers, the fewer risks they perceive, and the further developed better perception they have to follow COVID-19 preventive measures.

Additionally, the coefficients of purpose, which are also represented as R-Squared (R^2^), were assessed as a critical measure for measuring the endogenous latent variables of the structural model. R^2^ denotes the proportion of the variance of the dependent variable, which is expected from the independent variable, ranging between 0 and 1. Table 4 presents an R^2^ value of 0.317, denotes the predictors of ATT (TRGT and TRSM) interpret about 31.7% of its variance. Similarly, R^2^ 0.486 signifies the predictors of PBT (TRGT and TRSM) interpret about 48.6% of its variance, and 0.547 denotes the predictors of INT (TRGT, TRSM, ATT, PBT, PIIT, PRT), which interpret about 54.7% of its variance.

Moreover, Table 5, Table 6 and Table 7 report the structural model’s standardized direct, indirect, and total effects, respectively. As an example, the direct effect of TRSM on INT is 0.256, and its indirect impact is 0.116; therefore, the total effect of TRGT on INT is 0.372 = 0.256 + 0.116, that denotes, whereas TRSM increases by 1 standard deviation, and INT increases by 0.372 of a standard deviation.

#### 4.2.3. Moderating Investigation

The moderating investigation includes nine multiple regression models. In the multi-regression model, citizens’ readiness toward COVID-19 preventive measures is the dependent variable. Independent variables comprise HID, TRGT, TRSM, ATT, PBT, and the interaction term (HID × TRGT or HID × TRSM or HID × ATT or HID × PBT). The current study considers gender, age, and education as control variables because citizens’ demographic traits influence their readiness toward COVID-19 preventive measures.

Table 8 reports the findings of unstandardized regression coefficients and standard errors. Unstandardized regression coefficients specify the individual influence of X (independent variables) on Y (dependent variable). For example, in model 1, the regression coefficient 0.099 determines a change of 1 unit in the gender is related to a change of 0.099 units in the result of INT. As shown in Table 8, model 1, model 2, and model 3 are utilized to test the moderating effect of HID on the relationship between TRGT and citizens’ readiness toward COVID-19 preventive measures. The regression coefficient of the interaction term (HID×TRGT) is significant (*p* < 0.05), which denotes there is a moderating effect of health infodemic on the relationship between TRGT and citizen’s readiness toward COVID-19 preventive measures. Thus, H11a is supported.

Likewise, model 1, model 4, and model 5 are used to assess the moderating effect of HID on the relationship between TRSM and citizens’ readiness toward COVID-19 preventive measures. As Table 8 reports, the regression coefficient of the interaction term (HID×TRSM) is significant (*p* < 0.05), which denotes there is a moderating effect of HID on the relationship between TRSM and citizens’ readiness toward COVID-19 preventive measures. Thus, H11b is supported. 

As shown in Table 8, model 1, model 6, and model 7 are utilized to assess the moderating effect of HID on the relationship between ATT and citizens’ readiness toward COVID-19 preventive measures. The regression coefficient of the interaction term (HID×ATT) is significant (*p* < 0.05), which suggests there is a moderating effect of health infodemic on the relationship between ATT and citizens’ readiness toward COVID-19 preventive measures. Thus, H11c is supported.

Likewise, model 1, model 8, and model 9 are utilized to test the moderating effect of HID on the relationship between PBT and citizens’ readiness toward COVID-19 preventive measures. As Table 8 reports, the regression coefficient of the interaction term (HID×PBT) is significant (*p* < 0.05), which suggests there is a moderating effect of HID on the relationship between PBT and citizens’ readiness toward COVID-19 preventive measures. Thus, H11d is supported. This finding further indicates that HID deteriorates the predictive power of TRGT, TRSM, ATT, and PBT in determining citizens’ readiness toward COVID-19 preventive measures.

The moderating effect of health infodemics between TRGT, TRSM, ATT, and PBT, and citizens’ readiness toward COVID-19 preventive measures are depicted in Figure 6, Figure 7, Figure 8 and Figure 9. The perception of infodemics is a significant predictor of citizens’ readiness toward COVID-19 preventive measures, which negatively moderates the relationship between attitude, trust, perceived benefit, and perceived infodemic on citizens’ readiness toward COVID-19 preventive measures. TRGT and INT are different between citizens with high-infodemic perception and low-infodemic perception. Infodemic perception from low to high declines the influence of TRGT, TRSM, ATT, and PBT on citizens’ readiness toward COVID-19 preventive measures, which suggests there is the concurrent influence of attitude, trust, and perceived benefit and perceived infodemics on citizens’ readiness toward COVID-19 preventive measures.

Simultaneously, the regression investigation also specifies that TRGT has a considerable and positive influence on citizens’ readiness toward COVID-19 preventive measures. The perception of health infodemics has a significant and negative impact on citizens’ readiness toward COVID-19 preventive measures. Thus, the regression analysis supports H1. The perceived infodemic negatively lessens citizens’ readiness toward COVID-19 preventive measures. The finding of the regression investigation is inconsistent with those of the structural equation model. The potential cause is that the relationships between the constructs in SEM are more complex. These impact each other, which may mean that the findings are statistically insignificant; that is, differentiation in the investigation induces differentiation in conclusions. The current study’s findings align with previous studies [66] and consider that trust in government positively improves citizens’ readiness toward COVID-19 preventive measures.

Moreover, the current study explores meaningful inferences concerning the influence of demographic variables on citizens’ readiness. In comparison with women, men are keener to follow COVID-19 preventive measures. The effect of educational level and age are un-balanced while including other factors. The use of social media has a positive influence on citizens’ readiness. It indicates that citizens who spent a long time in social media show positive readiness toward COVID-19 preventive measures. The finding is consistent with the literature [30]. This suggests that a positive attitude and innovativeness play a vital role in shaping citizens’ willingness and readiness.

## 5. Discussion and Conclusions

### 5.1. Discussion

The current study develops an integrated research model to investigate how citizens’ trust in government and social media influence their readiness to follow COVID-19 pre-ventive measures. Additionally, we evaluate the moderating effect of “infodemic” on citi-zens’ motivation to follow COVID-19 preventive measures taken by the Taiwanese gov-ernment. The structural equation model (SEM) and multiple linear regression analysis were used to explore the proposed hypotheses. Findings indicate that citizens’ trust in social media, attitude, perceived benefit, peer referent, and personal innovativeness are signifi-cant predictors to evaluate citizens’ readiness to follow COVID-19 preventive measures. 

The current study finding indicates the relationship between citizens’ trust in the government and their readiness toward COVID-19 preventive measures is insignificant; thus, H1 is not supported. Citizens’ trust in government could be considered their satisfaction with the government’s services or information. Additionally, citizens’ greater trust in the government generally reduces their risk perception and improves the perception of benefits, resulting in a willingness to adopt new regulations. Due to the sudden implementation of strict lockdown, citizens were not psychologically and physically prepared to cope with the new regulations. They believed the government should inform them earlier. Citizens could display distrust and reserve critical determination towards services and the information provided by the government. As it further deteriorates citizens’ willingness to follow COVID-19 preventive measures. This assertion validates the result of statistical analysis and represents a negative cause-and-effect association between citizens’ TRGT and their readiness toward COVID-19 preventive measures. 

The relationship between TRSM and citizens’ readiness toward COVID-19 preventive measures is positive and significant, in line with previous studies [42,67]; thus, H2 is supported. Citizens who are better informed and confident in using social media attain the information to use it more positively and confidently, which further influences their level of satisfaction. Citizens’ satisfaction with using social media undoubtedly influences their perceptions of social media’s reliability, compassion, and integrity dimensions and makes them more eager to follow COVID-19 preventive measures. 

The current study finding explores attitude is positively related to citizens’ willingness to follow COVID-19 preventive measures, thus supporting H3. The current study finding shows that if citizens perceive a positive attitude toward COVID-19 preventive measures, their motivation to follow them improves. Therefore, citizens’ positive attitude should be measured as a significant determining factor, whereas citizens’ readiness toward COVID-19 preventive measures is concerned. The current study also finds other variables: ATT, PTB, PIIT, and PRT, which are essential for citizens’ readiness to follow COVID-19 preventive measures. 

The current study finds that perceived benefit significantly influences citizens’ readiness toward COVID-19 preventive measures and aligns with the findings of Yi et al. [68] and Hong et al. [54]. Perceived benefit generally indicates a better level of outcome. Therefore, positive perceived benefit suggests that citizens are ready to follow COVID-19 preventive measures, further reducing their risk perception regarding the COVID-19 pandemic. Thus, the government must provide better strategies and information through social media to further advance citizens’ readiness to follow COVID-19 preventive measures. 

Personal innovativeness proves to be a critical determinant of citizens’ readiness toward COVID-19 preventive measures, and the finding aligns with the conclusion of Rana et al. [69]. The following COVID-19 preventive measures are experiencing more frequent changes in citizens’ healthcare behavior than other behavioral changes in the healthcare context. It reasonably indicates better curiosity and confidence of citizens. As it frequently exhibits more inventive citizens ready to follow COVID-19 preventive measures. Citizens with higher personal innovativeness are inclined to accept a stable change and thus are more eager to follow it. However, such long-term inspiration of personal innovativeness has wide-ranging, feasible significance in citizens’ readiness to follow COVID-19 preventive measures. This finding aligns with Sun’s [13] results, indicating determined personal innovativeness of citizens allows them to pursue sustainable preventive measures of the COVID-19 pandemic. 

As expected, peer referents are found to influence citizens’ readiness toward COVID-19 preventive measures positively, is consistent with the previous studies, indicating that citizens’ attitude was a critical condition to the growth and development of positive readiness [70]. One potential explanation could be citizens under peer influence attempt to follow the expectations of others. Peer referents may be extrinsic motivational factors that inspire citizens to self-regulate COVID-19 preventive measures. The preventive measures of COVID-19 are implemented abruptly; thus, citizens are still in the developing phase. Therefore, improving normative views will improve citizens’ peer referents and enhance their readiness toward COVID-19 preventive measures.

Significant moderating results are explored between the relationship of health infodemics (HID) and TRGT, TRSM, ATT, PBT. 

The current study explores the negative influence of infodemics between citizens’ trust in social media and their readiness to follow COVID-19 preventive measures. It recommends that social media severely influence citizens’ behaviors concerning health beliefs. Primarily, health infodemic threads reduce the effectiveness of citizens’ performance, significance, and fraternity controls concerning healthcare behavior. In line with a previous study, this finding is in keeping with the result of the previous research, which demonstrated that the efficiency of citizens’ healthcare behavior is restrained due to infodemics [71]. Risk perception due to infodemics may intensify behavioral complications and complexity that diminish governmental controls’ efficiency in preventive measures. Therefore, social media can control the distribution of factual information by eliminating fact-checking and theoretically detrimental information. 

HID significantly moderates the relationships between attitude and citizens’ readiness to follow COVID-19 preventive measures and perceived benefit and citizens’ readiness to follow COVID-19 preventive measures. Rather than being a determinant of citizens’ readiness to follow COVID-19 preventive measures, HID has a relatively varying modifier power on the citizens’ readiness, explicitly practical inspirations. The result primarily explores citizens’ readiness to follow COVID-19 preventive measures. That is influenced by the health infodemic if citizens’ attitude and perception of benefit leading to follow COVID-19 preventive measures are useful. The interference of the relationship between attitude and COVID-19 preventive measures by HID reduced citizens’ readiness. Based on the findings, we interpret that perceived negative impression of HID mainly influences the attitude, which is primarily dominated by emotional encouragement, satisfaction, and authority, as harming further or even damaging citizens’ readiness to follow COVID-19 preventive measures. However, attitude is debarred by HID to stop citizens from following COVID-19 preventive measures, which has been initially decided to be followed.

### 5.2. Theoretical Implications

The current study explores the influence of citizens’ attitude, trust, and perceived benefits on their readiness to follow COVID-19 preventive measures from a theoretical context. Although attitude has received significant consideration concerning adopting innovative healthcare behavior, it has been rarely explored in citizens’ readiness to follow the COVID-19 preventive measures. That requires citizens’ highest level of attitudinal readiness as they need to switch entirely to a new form of behavioral method. Thus, citizens needed psychological readiness to overcome the uncertain consequences. Additionally, the literature explored online system user behavior [13,48,54] from the perspective of information technology adoption theories, TAM, IDT, and UTAUT. It has rarely explored the effect of citizens’ psychological readiness to follow new preventive measures. As citizens are ready to follow COVID-19 preventive measures during the pandemic, demand for developing a sustainable health behavioral program has been considered. Therefore, it is required to identify comprehensive factors influencing citizens’ readiness to follow COVID-19 preventive measures.

Kim et al. [20] stated that publicizing correct information would positively supervise the present century’s highest public health adversity. It could make reasonable improvements in making decisions based on information and misinformation. Thus, it would be more appropriate to inspire citizens to evaluate the information’s credibility before believing or sharing it. Cinelli et al. [24] recommended that sharing misinformation concerning healthcare could have a significant consequence, especially during a pandemic. Situational inspiration can improve citizens’ participation in confirming the information with the reputed organizations such as WHO, CDC, etc., that ultimately influences citizens’ concerns.

The proposed research model incorporates trust, attitude, and perceived benefit and evaluates their relationship with citizens’ intention to follow COVI-19 preventive measures. Based on the authors’ knowledge, the current study is the first study that developed and analyzed moderating effect of health infodemics between trust, attitude, and perceived benefit and citizens’ readiness to follow COVID-19 preventive measures. From a theoretical perspective, the current study’s findings support academicians understanding of how citizens’ different degrees of health awareness influence their trust, attitude, and perception of the benefit of following a new healthcare behavior.

### 5.3. Practical Implications

Based on the current study’s findings, we suggest some vital ideas to public health authorities in Taiwan and other governments. 

Trust in the government and social media are crucial factors that positively develop citizens’ readiness to follow COVID-19 preventive measures. These factors also influence attitude, perception of benefits, and adoption intention of COVID-19 preventive measures. The government should relentlessly develop the administration’s credibility and preserve social media reliability. Additionally, the government should initiate strategies to boost citizens’ digital abilities as society is transforming into digital—especially those old and uncomfortable with new technologies.

Subsequently, the government of a country and international organizations such as the World health organization (WHO), United Nations (UN), United Nations Children’s Emergency Fund (UNICEF) independently and cooperatively should organize some scientific-educational learning accommodations for local organizations, health officials of the government, and community legislatures. The government should also take the initiative and inform social media providers to be more proficient and uncompromising concerning the vulnerable distribution of catastrophic during the pandemic. In the same way, social media providers could make users aware of the negative influence of sharing information without confirmation, specifically while it is associated with the healthcare of mass.

The citizens’ readiness to follow COVID-19 preventive measures could be affected by peer referents such as family members, networks, and social media. Hereafter, the government should use social media, notice boards, correspondents, television, and transistors to make citizens aware of what to do or not during the COVID-19 pandemic.

### 5.4. Limitations and Future Studies

Despite its significant conclusions and implications, the current study experiences some limitations. First, inferences are from single learning samples collected from Taiwan. Thus, researchers must be careful while generalizing the findings with other healthcare behavior perspectives. Future studies should examine a multicultural perspective to examine factors influencing citizens’ readiness to follow COVID-19 preventive measures. Second, the current study samples do not demonstrate all citizens groups because of the restrictions of interval and space. Thus, future research should include different ethnic groups, improving the current study’s representativeness.

## Figures and Tables

**Figure 1 ijerph-19-01347-f001:**
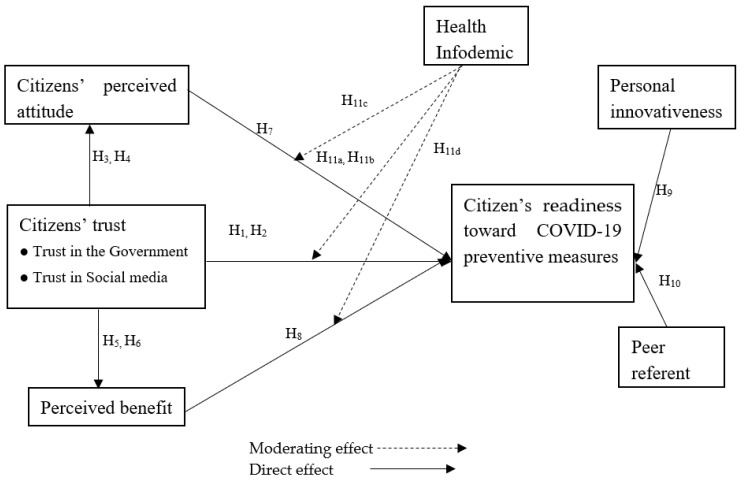
Research model.

**Figure 2 ijerph-19-01347-f002:**
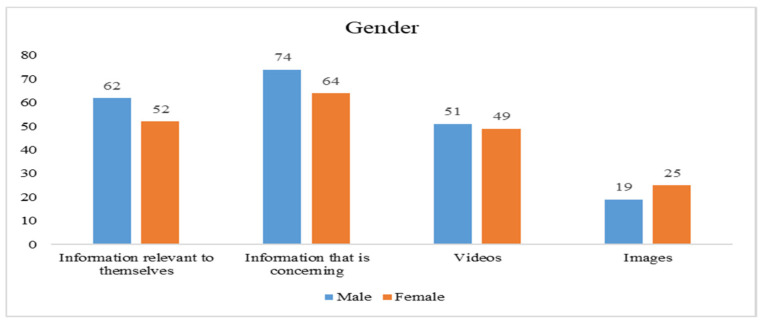
Gender distribution of the study respondents.

**Figure 3 ijerph-19-01347-f003:**
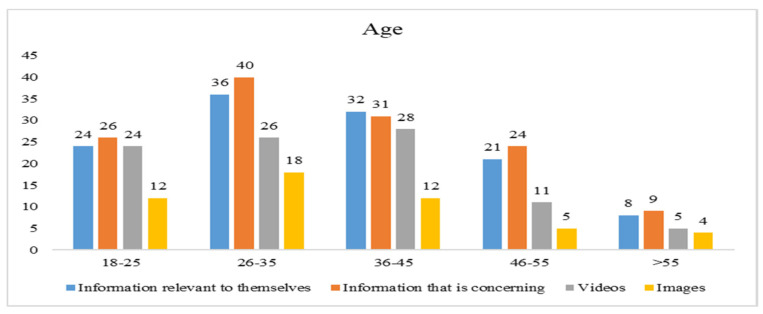
Age distribution of the study respondents.

**Figure 4 ijerph-19-01347-f004:**
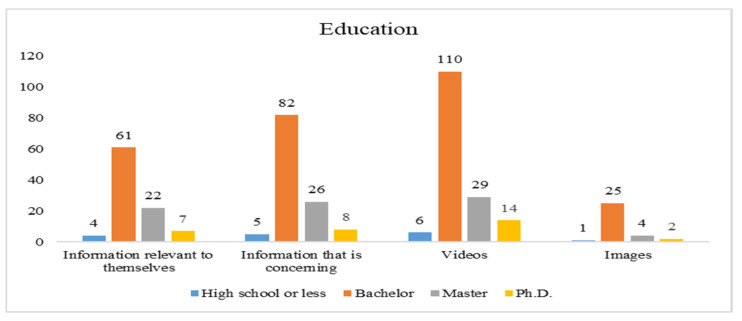
Educational distribution of the study respondents.

**Figure 5 ijerph-19-01347-f005:**
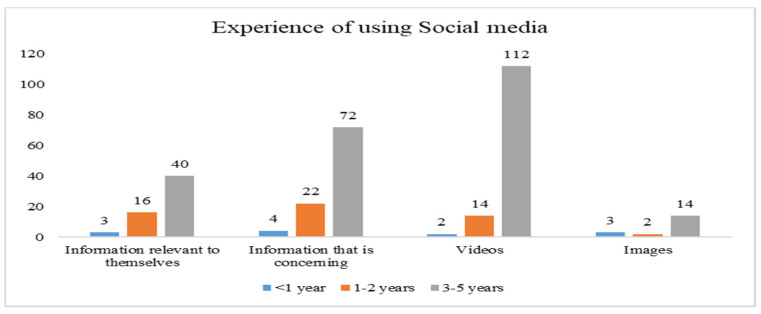
Experience of social media of the study respondents.

**Figure 6 ijerph-19-01347-f006:**
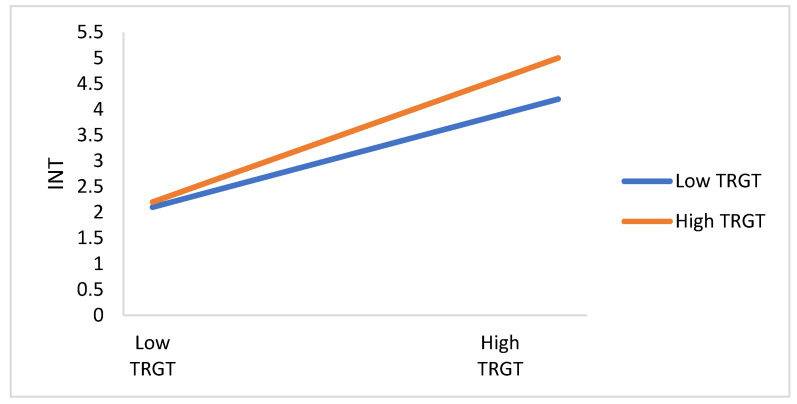
The plot of the moderating effect of health infodemics on the relationship between TRGT and INT.

**Figure 7 ijerph-19-01347-f007:**
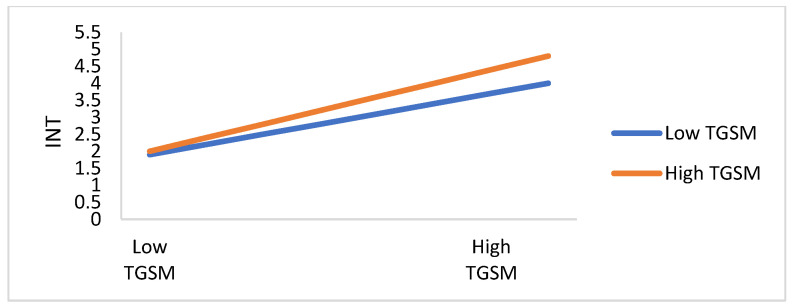
The plot of the moderating effect of health infodemics on the relationship between TRSM and INT.

**Figure 8 ijerph-19-01347-f008:**
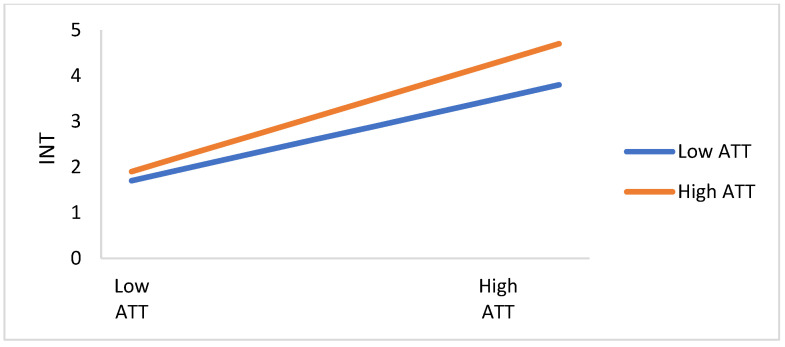
The plot of the moderating effect of health infodemics on the relationship between ATT and INT.

**Figure 9 ijerph-19-01347-f009:**
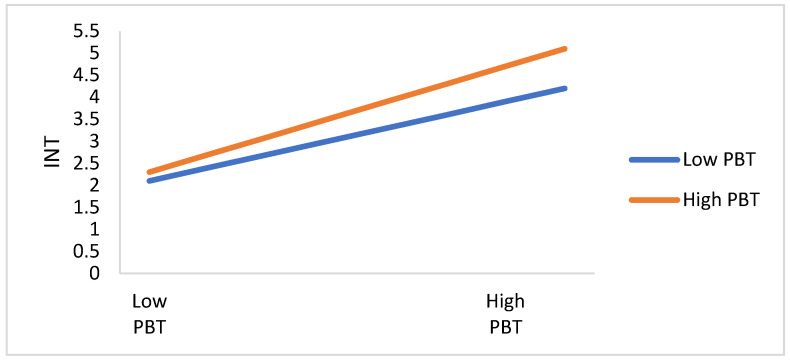
The plot of the moderating effect of health infodemics on the relationship between PBT and INT.

**Table 1 ijerph-19-01347-t001:** Cronbach’s alpha value and Pearson correlation.

	ATT	TRGT	TRSM	PBT	PIIT	PRT	HID	INT
ATT	1							
TRGT	0.265 **	1						
TRSM	0.412 **	−0.235 **	1					
PBT	0.312 **	−0.318 **	0.430 **	1				
PIIT	0.321 **	−0.187 **	0.260 **	0.432 **	1			
PRT	0.124 **	0.486 **	−0.176 **	−0.217 **	−0.249 **	1		
HID	−0.187 **	−0.156 **	−0.425 **	−0.459 **	−0.458 **	−0.217 **	1	
INT	0.246 **	0.272 **	0.321 **	0.421 **	0.287 **	0.318 **	0.417 **	1
Mean	3.712	3.215	3.427	3.518	3.316	3.628	3.845	3.582
S.D.	0.928	0.956	0.967	1.000	0.868	0.891	0.948	0.986

** Correlation is significant at the 0.01 level (two-tailed).

**Table 2 ijerph-19-01347-t002:** Results of reliability and validity test.

Construct.	Code	Cronbach’s α	Factor Loadings	CR	AVE
Attitude	ATT1	0.841	0.865	0.812	0.816
	ATT2		0.923		
	ATT3		0.905		
Trust on the Government	TRGT1	0.926	0.881	0.891	0.646
	TRGT2		0.941		
	TRGT3		0.913		
Trust in Social media	TRSM1	0.852	0.856	0.686	0.756
	TRSM2		0.841		
	TRSM3		0.876		
Perceived benefit	PBT1	0.868	0.912	0.781	0.742
	PBT2		0.889		
	PBT3		0.891		
Personal innovativeness	PIIT1	0.872	0.912	0.824	0.708
	PIIT2		0.872		
	PIIT3		0.946		
Peer referent	PRT1	0.787	0.932	0.746	0.684
	PRT2		0.856		
	PRT3		0.862		
Health infodemic	HID1	0.792	0.878	0.821	0.748
	HID2		0.892		
	HID3		0.849		
	HID4		0.870		
	HID5		0.858		
Readiness toward COVID-19 preventive measures	INT1	0.897	0.907	0.868	0.657
	INT2		0.895		
	INT3		0.916		
	INT4		0.894		

**Table 3 ijerph-19-01347-t003:** Path coefficients of structural equation model.

		Path Coefficient	C.R.	*p*-Value	Result
H1	TRGT→INT	−0.026	−0.516	0.716	Rejected
H2	TRSM→INT	0.252 **	3.156	0.001	Supported
H3	TRGT→ATT	0.217 **	3.126	0.014	Supported
H4	TRSM→ATT	0.116 **	2.635	0.007	Supported
H5	TRGT→PBT	0.256 ***	4.846	0.000	Supported
H6	TRSM→PBT	0.340 ***	6.642	0.000	Supported
H7	ATT→INT	0.120 **	3.178	0.001	Supported
H8	PBT→INT	0.264 ***	5.662	0.000	Supported
H9	PIIT→INT	0.418 ***	8.217	0.000	Supported
H10	PRT→INT	0.284 ***	5.517	0.000	Supported

Note: ** *p* < 0.01, and *** *p* < 0.001.

**Table 4 ijerph-19-01347-t004:** Coefficients of determination.

Construct	R^2^
ATT	0.317
PBT	0.486
INT	0.547

**Table 5 ijerph-19-01347-t005:** Direct effect in the structural model.

	TRGT	TRSM	ATT	PBT	PIIT	PRT
ATT	0.116	0.256				
PBT	0.120	0.264				
INT	−0.026	0.252	0.217	0.340	0.418	0.284

**Table 6 ijerph-19-01347-t006:** Indirect effect in the structural model.

	TRGT	TRSM	ATT	PBT	PIIT	PRT
ATT						
PBT						
INT	0.089	0.116				

**Table 7 ijerph-19-01347-t007:** Total effect in the structural model.

	TRGT	TRSM	ATT	PBT	PIIT	PRT
ATT	0.116	0.256				
PBT	0.120	0.264				
INT	0.063	0.372	0.217	0.340	0.418	0.284

**Table 8 ijerph-19-01347-t008:** The effect of TRGT, TRSM, ATT, and PBT on INT is moderated by health infodemic.

	Model 1	Model 2	Model 3	Model 4	Model 5	Model 6	Model 7	Model 8	Model 9
Gender	0.099 * (0.05)	0.101 * (0.046)	0.097 * (0.046)	0.113 * (0.047)	0.107 * (0.047)	0.113 * (0.049)	0.095 * (0.049)	0.097 * (0.048)	0.115 * (0.048)
Age	−0.041 * (0.026)	−0.063 * (0.025)	−0.062 * (0.024)	−0.045 * (0.024)	−0.045 (0.024)	−0.061 * (0.023)	−0.064 * (0.026)	−0.043 * (0.026)	−0.043 (0.026)
Education	0.031 (0.028)	0.031 * (0.027)	0.034 (0.027)	0.038 * (0.027)	0.040 * (0.027)	0.042 * (0.029)	0.044 (0.029)	0.047 * (0.029)	0.050 * (0.029)
Experience of using Social media	0.216 *** (0.037)	0.224 *** (0.035)	0.221 *** (0.035)	0.218 *** (0.035)	0.219 *** (0.035)	0.217 *** (0.035)	0.226 *** (0.035)	0.223 *** (0.035)	0.220 *** (0.035)
HID		−0.096 *** (0.028)	−0.092 *** (0.028)	−0.063 * (0.029)	−0.065 * (0.029)	−0.094 *** (0.030)	−0.090 *** (0.030)	−0.061 * (0.031)	−0.063 * (0.031)
TRGT		0.242 *** (0.029)	0.25796 *** (0.030)						
HID × TRGT			−0.062 * (0.024)						
TRSM				0.269 *** (0.032)	0.269 *** (0.032)				
HID × TRSM					0.041 * (0.022)				
ATT						0.277 *** (0.034)	0.277 *** (0.034)		
HID × ATT							−0.039 * (0.021)		
PBT								0.289 *** (0.036)	0.289 *** (0.036)
HID × PBT									−0.039 * (0.019)
R^2^	0.068	0.162	0.168	0.172	0.175	0.182	0.188	0.192	0.195
Adjusted R2	0.064	0.157	0.162	0.166	0.170	0.177	0.182	0.187	0.190
∆R^2^	0.068 ***	0.115 ***	0.03 *	0.124 ***	0.005 *	0.119 ***	0.005 *	0.128 ***	0.03 *

Note: * *p* < 0.05 and *** *p* < 0.001.

## Data Availability

The source of the items is stated in Appendix A.

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
