# Peer review of "Role of Infodemics on Social Media in the Development of People’s Readiness to Follow COVID-19 Preventive Measures"

_ijerph, 2022, doi:10.3390/ijerph19031347_

Round 1

Reviewer 1 Report

I am sensitive to how English is likely not the authors' first language. However, extensive English language editing is required. I am unable to understand numerous large sections of the manuscript, and therefore I cannot provide a quality review. 

Author Response

  1. I am sensitive to how English is likely not the authors' first language. However, extensive English language editing is required. I am unable to understand numerous large sections of the manuscript, and therefore I cannot provide a quality review.

Ans. Thank you for your recommendation. We rectify the English, which will be found throughout the paper.

This misinformation traverses borders with…intensifies faster and further. (Line 14-15)

Currently, several countries have started…the body to fight an infection [4]. (Line 43-44)

Different types of vaccines such as…immune system to respond to an invader. (Line 44-47)

Still, each differs in how it… trains the body to fight this invader [5]. (Line 47-48)

Taiwan's government dynamically accelerated…outbreak in April 2021 [7]. (Line 56-58)

The Ministry of Education, the Ministry…and fully vaccinated by Jan. 1, 2022. (Line 58-61)

78.74% of the population received two doses, and…dose of vaccine till now [7]. (Line 62-63)

Taiwan’s Government also…tests irrespective of their vaccination status [7]. (Line 63-65) Social media has emerged as …awareness and attitude towards specific issues. (Line 78-80) Misinformation or infodemic is stated…or unintentionally) on social media [9] (Line 83-85)

Thus, the current study incorporates…follow COVID-19 preventive measures. (Line 96-98)

The purpose of the current study…measures during the COVID-19 pandemic. (Line 98-100)

We employ two factors based on existing… a construct called citizens' trust. (Line 100-101)

Additionally, citizens' attitude and perceived…preventive measures. (Line 101-103) Misinformation was generally pervasive…such as the HIV epidemic. (Line 112-113)

It, too, was beleaguered by conspiracy… still visible in areas to this day [21]. (Line 113-114)

This mainly was empha-sized by the…effectiveness of HIV medication [21]. (Line 114-117)

As an alternative, the government promoted… more than 300,000 lives [21]. (Line 117-120) As COVID-19 turns into a full-fledged public…fast on the internet [21,22]. (Line 121-122)

Despite scientists from several countries… and following on the inter-net. (Line 122-124)

Moreover, the high rate of COVID-19 …undeniably erudite much [22]. (Line 124-126)

However, there is a concern that such a…degrees of misinformation [22]. (Line 126-129)

With mounting digitization, the development…broadcast more quickly [23]. (Line 130-131)

This could assist in more rapidly…intensify detrimental information [22,23]. (Line 131-132)

An infodemic regarding COVID-19 can…the wellbeing of nearby people [21]. (Line 132-135) Misinformation has spread far and wide, …sources of infor-mation [23]. (Line 136-137)

Uncredited sources have muddled basic …exposure to the virus [22]. (Line 137-138)

The public health crisis emerging due to…feel misinformation's effects [21]. (Line 138-139)

It creates misperception and risk-taking…improve detriment well-being [24]. (Line 139-140)

The spread of false information drowns…of virus transmission [24]. (Line 140-143)

It also distrusts health authorities and… demoralizes the citizens' health retort. (Line 143)

The influence of these false arguments…has the potential to be fatal [21,23]. (Line 153-155)

In the face of a pandemic, governments… a public health crisis [22,24,25]. (Line 155-158)

Additionally, it is dangerous for… governments to politicize this pandemic. (Line 158) Algorithms intermediate and simplify… and information distribution [27]. (Line 172-173) Users are inclined to obtain information…around collective narrations [34]. (Line 176-178) Literature explored citizens' trust in government… perceived benefit [45,47]. (Line 232-233) Dutta et al. [48] stated attitude as individual…information to specific subjects. (Line 240-241)

In their study on adopting healthcare…an individual's behavioral intention. (Line 241-243)

Gong et al. [52] explored that individuals…accepting well-being behaviors. (Line 253-254)

H9. Personal innovativeness positively…COVID-19 preventive measures. (Line 276-277)

H10. Peer referent positively affects…follow COVID-19 preventive measures. (Line 291-292)

Thus, the current study theorizes…toward COVID-19 preventive measures. (Line 302-304)

Thus, the current study considers health… COVID-19 preventive measures. (Line 309-311)

The current study used ... develop and validate the proposed study model. (Line 336-337)

The final questionnaire consists…and demographic information of responders. (Line 349-350)

The Delphi method was used to validate… the initial conceptual framework. (Line 367)

Four experts formed an expert …working as an Instructor in a university. (Line 367-369)

All of the panelists have more than 12 years…the Information science area. (Line 369-370)

Another panelist works in a hospital with… health informatics associate. (Line 370-371)

Three experts have received their…to 40-50 and one belongs to 45-50). (Line 372-374)

The significant concern to ask those experts…to the information science. (Line 374-376)

After two rounds of expert panel meetings…proposed for a pilot study. (Line 376-377)

We use a 5-point Likert scale, with a measure… agree, and strongly agree. (Line 377-378)

Each round of expert panel meetings… and relevant literature reviews. (Line 378-381)

After receiving the questionnaire from experts…checked, respectively. (Line 382-383)

The Chronbach’s α values in the first … COVID-19 preventive measures. (Line 383-387)

Furthermore, two items were deleted…for round two of the Delphi method. (Line 387-388)

After the second round, Chronbach’s α value…deviation between 0.5 to 1.5. (Line 389-391)

Based on the suggestions from the…the proposed initial research framework. (Line 391-393)

However, items were increased from…government and trust in social media. (Line 393-395)

Researchers currently use several survey…  negative, or neutral ways [59]. (Line 397-401)

However, we used the convenience… of using a randomized sample [61]. (Line 401-404)

Nine responses were unable to be …complete responses, missing data, etc. (Line 409-410)

Figures 2, 3, 4, and 5 indicate how…of using social media, respectively. (Line 411-413)

According to the national data of Taiwan…is between 15 to 64 years [62]. (Line 413-415)

However, the ratio of males and…sample is 52% and 48%, respectively. (Line 415-416)

The majority of respondents (78%) were in the… had a college degree. (Line 416-418)

These findings point out that all related …using SEM and regression. (Line 451-452)

In the current study, several practical… and statistical analyses were used. (Line 455)

As Table 2 confirms, the constructs' AVE values…for convergent validity. (Line 480-482)

Citizens' higher innovativeness indicates…and are prepared to follow them. (Line 500-502)

Moreover, Tables 5–7 report the structural…and total effects, respectively. (Line 517-518)

Table 8 reports the findings with…regression coefficients and standard errors. (Line 532-533)

The potential cause is that the relationships …in SEM are more complex. (Line 588-589)

They impact each other, which might cause…differentiation in conclusions. (Line 589-591)

The current study's findings are in…to-ward COVID-19 preventive measures. (Line 591-594)

But the relationship between TRSM… in line with previous studies [42,67]. (Line 625-626)

The current study finding shows that…motivation to follow them improves. (Line 636-638)

The current study explores how the… of Yi et al. [68] and Hong et al. [54]. (Line 641-643)

The current study finding explores… respective healthcare behaviors. (Line 677-679)

HID was found to significantly moderate…COVID-19 preventive measures. (Line 690-692)

Based on the findings, we interpret…follow COVID-19 preventive measures. (Line 699-702)

The proposed research model incorporates…and evaluates their relationship. (Line 731-732)

In theory, it helps academicians understand…following new health behavior. (Line 735-737)

Trust in the government and social media…COVID-19 preventive measures. (Line 741-742)

Additionally, facing the digital transformation…with new technologies. (Line 745-747)

Reviewer 2 Report

I want to congratulate the Authors for the work they have done and the article. It is very well prepared and written. However, nevertheless, I have some questions and comments.

1. line 45 should be supplemented with a short paragraph on attempts to find a therapy for COVID-19 (both by using already known drugs in therapy and developing entirely new ones).
2. The Introduction should also briefly mention that depending on social, geographical and cultural conditions - each country had its own specific approach to the pandemic and slightly different solutions. In this context, it is worth emphasizing the specificity of Taiwan. The authors are aware of this. They mention it in line 670. Therefore, it is worth extending this thread a bit.

3 In paragraph 2.1. (Health Infodemics) I would also suggest adding a thread about the low credibility of scientific publications on COVID-19. I would suggest focusing solely on this issue. It is already quite well described even in severe scientific literature, for example:
https://bmcmedicine.biomedcentral.com/articles/10.1186/s12916-020-01556-3
https://www.tandfonline.com/doi/abs/10.1080/08989621.2020.1782203
https://www.biotechniques.com/covid-19/repif_filtering-out-unreliable-covid-19-research/
https://www.sciencemag.org/news/2020/06/two-elite-medical-journals-retract-coronavirus-papers-over-data-integrity-questions

plus popular science publications:
https://www.smithsonianmag.com/science-nature/how-avoid-misinformation-about-covid-19-180974615/
https://www.economist.com/science-and-technology/2020/05/07/scientific-research-on-the-coronavirus-is-being-released-in-a-torrent
https://www.insidehighered.com/news/2020/06/29/new-mit-press-journal-debunk-bad-covid-19-research
https://www.insidehighered.com/news/2020/06/08/fast-pace-scientific-publishing-covid-comes-problems
http://www.rationaloptimist.com/blog/what-the-pandemic-has-taught-us/
https://news.sky.com/story/coronavirus-dr-johnny-bananas-and-dr-person-fakename-among-medical-signatories-on-herd-immunity-open-letter-12099947

In section 3.2, the information on the convenience sampling method is too perfunctory and needs to be described in more detail. First, in the demographic structure, the 18-45 age group is overrepresented (how much does it differ from the real structure of Taiwanese society?)

The survey method section is worth mentioning other methods such as sentiment analysis based on Big Data technologies.

Author Response

I want to congratulate the Authors for the work they have done and the article. It is very well prepared and written. However, nevertheless, I have some questions and comments.

  1. line 45 should be supplemented with a short paragraph on attempts to find a therapy for COVID-19 (both by using already known drugs in therapy and developing entirely new ones).

Ans. Thank you for your recommendation. We add 3-5 lines, which will be found in line 43-48.

Currently, several countries have started…the body to fight an infection [4]. (Line 43-44)

Different types of vaccines such as…immune system to respond to an invader. (Line 44-47)

Still, each differs in how it… trains the body to fight this invader [5]. (Line 47-48)

  1. The Introduction should also briefly mention that depending on social, geographical and cultural conditions - each country had its own specific approach to the pandemic and slightly different solutions. In this context, it is worth emphasizing the specificity of Taiwan. The authors are aware of this. They mention it in line 670.Therefore, it is worth extending this thread a bit.

Ans. Thank you for your recommendation. We include the strategies taken by the Taiwan government, which will be found in line 56-65.

Taiwan's government dynamically accelerated…outbreak in April 2021 [7]. (Line 56-58)

The Ministry of Education, the Ministry…and fully vaccinated by Jan. 1, 2022. (Line 58-61)

78.74% of the population received two doses, and…dose of vaccine till now [7]. (Line 62-63)

Taiwan’s Government also…tests irrespective of their vaccination status [7]. (Line 63-65)

  1. In paragraph 2.1. (Health Infodemics) I would also suggest adding a thread about the low credibility of scientific publications on COVID-19. I would suggest focusing solely on this issue. It is already quite well described even in severe scientific literature, for example:

https://bmcmedicine.biomedcentral.com/articles/10.1186/s12916-020-01556-3

https://www.tandfonline.com/doi/abs/10.1080/08989621.2020.1782203

https://www.biotechniques.com/covid-19/repif_filtering-out-unreliable-covid-19-research/

https://www.sciencemag.org/news/2020/06/two-elite-medical-journals-retract-coronavirus-papers-over-data-integrity-questions

plus popular science publications:

https://www.smithsonianmag.com/science-nature/how-avoid-misinformation-about-covid-19-180974615/

https://www.economist.com/science-and-technology/2020/05/07/scientific-research-on-the-coronavirus-is-being-released-in-a-torrent

https://www.insidehighered.com/news/2020/06/29/new-mit-press-journal-debunk-bad-covid-19-research

https://www.insidehighered.com/news/2020/06/08/fast-pace-scientific-publishing-covid-comes-problems

http://www.rationaloptimist.com/blog/what-the-pandemic-has-taught-us/

https://news.sky.com/story/coronavirus-dr-johnny-bananas-and-dr-person-fakename-among-medical-signatories-on-herd-immunity-open-letter-12099947

Ans. Thank you for your recommendation. We modify the part Health infodemic according to reviewer’s recommendation, which will be found in line 112-158.

Misinformation was generally pervasive…such as the HIV epidemic. (Line 112-113)

It, too, was beleaguered by conspiracy… still visible in areas to this day [21]. (Line 113-114)

This mainly was emphasized by the…effectiveness of HIV medication [21]. (Line 114-117)

As an alternative, the government promoted… more than 300,000 lives [21]. (Line 117-120)

As COVID-19 turns into a full-fledged public…fast on the internet [21,22]. (Line 121-122)

Despite scientists from several countries… and following on the internet. (Line 122-124)

Moreover, the high rate of COVID-19 …undeniably erudite much [22]. (Line 124-126)

However, there is a concern that such a…degrees of misinformation [22]. (Line 126-129)

With mounting digitization, the development…broadcast more quickly [23]. (Line 130-131)

This could assist in more rapidly…intensify detrimental information [22,23]. (Line 131-132)

An infodemic regarding COVID-19 can…the wellbeing of nearby people [21]. (Line 132-135) Misinformation has spread far and wide, …sources of information [23]. (Line 136-137)

Uncredited sources have muddled basic …exposure to the virus [22]. (Line 137-138)

The public health crisis emerging due to…feel misinformation's effects [21]. (Line 138-139)

It creates misperception and risk-taking…improve detriment well-being [24]. (Line 139-140)

The spread of false information drowns…of virus transmission [24]. (Line 140-143)

It also distrusts health authorities and… demoralizes the citizens' health retort. (Line 143)

The influence of these false arguments…has the potential to be fatal [21,23]. (Line 153-155)

In the face of a pandemic, governments… a public health crisis [22,24,25]. (Line 155-158)

Additionally, it is dangerous for… governments to politicize this pandemic. (Line 158)

  1. In section 3.2, the information on the convenience sampling method is too perfunctory and needs to be described in more detail. First, in the demographic structure, the 18-45 age group is over represented (how much does it differ from the real structure of Taiwanese society?)

Ans. Thank you for your recommendation. We describe other survey technologies such as Sentiment analysis and explain why age distribution of the current study is considered unbalanced, which will be found in line 397-404 and 411-418.

Researchers currently use several survey…  negative, or neutral ways [59]. (Line 397-401)

However, we used the convenience… of using a randomized sample [61]. (Line 401-404)

Figures 2, 3, 4, and 5 indicate how…of using social media, respectively. (Line 411-413)

According to the national data of Taiwan…is between 15 to 64 years [62]. (Line 413-415)

However, the ratio of males and…sample is 52% and 48%, respectively. (Line 415-416)

The majority of respondents (78%) were in the… had a college degree. (Line 416-418)

  1. 5. The survey method section is worth mentioning other methods such as sentiment analysis based on Big Data technologies.

Ans. Thank you for your recommendation. We describe other survey technologies such as Sentiment analysis, will be found in line 397-404.

Researchers currently use several survey…  negative, or neutral ways [59]. (Line 397-401)

However, we used the convenience… of using a randomized sample [61]. (Line 401-404)

Reviewer 3 Report

Congratulations to the authors for their research proposal, it is very comprehensive and well documented. However, I would suggest some revisions:
1) it would be appropriate to detail how the method was validated since they talk about business and academic experts but who they are and how it was done.
2) The design of the tables of results does not make it easy to understand, so I suggest another more accessible form of presentation, for example with graphics.

Author Response

Congratulations to the authors for their research proposal, it is very comprehensive and well documented. However, I would suggest some revisions:

1) it would be appropriate to detail how the method was validated since they talk about business and academic experts but who they are and how it was done.

Ans. Thank you for your recommendation. We include the section The Delphi Method to explain how the method was validated, will be found in line 366-395.

The Delphi method was used to validate… the initial conceptual framework. (Line 367)

Four experts formed an expert …working as an Instructor in a university. (Line 367-369)

All of the panelists have more than 12 years…the Information science area. (Line 369-370)

Another panelist works in a hospital with… health informatics associate. (Line 370-371)

Three experts have received their…to 40-50 and one belongs to 45-50). (Line 372-374)

The significant concern to ask those experts…to the information science. (Line 374-376)

After two rounds of expert panel meetings…proposed for a pilot study. (Line 376-377)

We use a 5-point Likert scale, with a measure… agree, and strongly agree. (Line 377-378)

Each round of expert panel meetings… and relevant literature reviews. (Line 378-381)

After receiving the questionnaire from experts…checked, respectively. (Line 382-383)

The Chronbach’s α values in the first … COVID-19 preventive measures. (Line 383-387)

Furthermore, two items were deleted…for round two of the Delphi method. (Line 387-388)

After the second round, Chronbach’s α value…deviation between 0.5 to 1.5. (Line 389-391)

Based on the suggestions from the…the proposed initial research framework. (Line 391-393)

However, items were increased from…government and trust in social media. (Line 393-395)

2) The design of the tables of results does not make it easy to understand, so I suggest another more accessible form of presentation, for example with graphics.

Ans. Thank you for your recommendation. We include Figures instead of Tables, which will be found in line 420-430.

Figures 2, 3, 4, and 5 indicate how…of using social media, respectively.

Reviewer 4 Report

Informative introduction presented. 

Line 323 - The target population is Taiwanese citizens. (missing the word citizen?).

Line 326 - Sentence needs fixing

Line 330 - Sentence needs fixing

The paper flows nicely, one thing I feel is missing in the literature review are  examples of people spreading false information. In many countries there are government officials, medical doctors, people with special authority spreading false information about the vacination and the virus. Some believe in it some dont. Perhaps bring out some concrete examples from your country or other places.

Besides that I am please with the methodology  selection and presentation. Good job and happy holidays.

Author Response

Informative introduction presented.

  1. Line 323 - The target population is Taiwanese citizens. (missing the word citizen?).

Ans. Thank you for your recommendation. We include the word citizens, will be found in line 397.

The target population for the current study was Taiwanese citizens. (Line 397)

  1. Line 326 - Sentence needs fixing

Ans. Thank you for your recommendation. We modify the sentence, will be found in line 404-405.

An online survey was conducted to collect data. (Line 404-405)

  1. Line 330 - Sentence needs fixing

Ans. Thank you for your recommendation. We modify the sentence, will be found in line 409-410.

Four hundred five responses were returned. (Line 409)

Nine responses were unable to be used due to incomplete responses, missing data, etc. (Line 409-410)

  1. The paper flows nicely, one thing I feel is missing in the literature review are examples of people spreading false information. In many countries there are government officials, medical doctors, people with special authority spreading false information about the vaccination and the virus. Some believe in it some dent. Perhaps bring out some concrete examples from your country or other places.

Ans. Thank you for your recommendation. We modify the section Health infodemic, will be found in line 112-158.

Misinformation was generally pervasive…such as the HIV epidemic. (Line 112-113)

It, too, was beleaguered by conspiracy… still visible in areas to this day [21]. (Line 113-114)

This mainly was empha-sized by the…effectiveness of HIV medication [21]. (Line 114-117)

As an alternative, the government promoted… more than 300,000 lives [21]. (Line 117-120)

As COVID-19 turns into a full-fledged public…fast on the internet [21,22]. (Line 121-122)

Despite scientists from several countries… and following on the internet. (Line 122-124)

Moreover, the high rate of COVID-19 …undeniably erudite much [22]. (Line 124-126)

However, there is a concern that such a…degrees of misinformation [22]. (Line 126-129)

With mounting digitization, the development…broadcast more quickly [23]. (Line 130-131)

This could assist in more rapidly…intensify detrimental information [22,23]. (Line 131-132)

An infodemic regarding COVID-19 can…the wellbeing of nearby people [21]. (Line 132-135)

Misinformation has spread far and wide, …sources of information [23]. (Line 136-137)

Uncredited sources have muddled basic …exposure to the virus [22]. (Line 137-138)

The public health crisis emerging due to…feel misinformation's effects [21]. (Line 138-139)

It creates misperception and risk-taking…improve detriment well-being [24]. (Line 139-140)

The spread of false information drowns…of virus transmission [24]. (Line 140-143)

It also distrusts health authorities and… demoralizes the citizens' health retort. (Line 143)

The influence of these false arguments…has the potential to be fatal [21,23]. (Line 153-155)

In the face of a pandemic, governments… a public health crisis [22,24,25]. (Line 155-158)

Additionally, it is dangerous for… governments to politicize this pandemic. (Line 158)

  1. Besides that, I am please with the methodology selection and presentation. Good job and happy holidays.

Ans. Thank you.

Round 2

Reviewer 1 Report

Although some improvements have been made, there remains numerous English language problems that make the manuscript difficult to follow. 

Author Response

Thank you for your comments, we modify the paper, and you will find it throughout the paper. 

Please go through the attached file
